# Structural basis of interprotein electron transfer in bacterial sulfite oxidation

Aaron P McGrath[1][†], Elise L Laming[2][‡], G Patricia Casas Garcia[3], Marc Kvansakul[3], J Mitchell Guss[2], Jill Trewhella[2], Benoit Calmes[4,5], Paul V Bernhardt[4,5], Graeme R Hanson[4,6][§], Ulrike Kappler[4,5]*, Megan J Maher[3]*

[1]Structural Biology Program, Centenary Institute, Sydney, Australia; [2]School of Molecular Bioscience, University of Sydney, Sydney, Australia; [3]Department of Biochemistry and Genetics, La Trobe Institute for Molecular Science, La Trobe University, Melbourne, Australia; [4]Centre for Metals in Biology, The University of Queensland, Brisbane, Australia; [5]School of Chemistry and Molecular Biosciences, The University of Queensland, Brisbane, Australia; [6]Centre for Advanced Imaging, University of Queensland, Brisbane, Australia

*For correspondence:
u.kappler1@uq.edu.au (UK);
m.maher@latrobe.edu.au (MJM)

Present address: [†] Skaggs School of Pharmacy and Pharmaceutical Sciences, University of California, San Diego, La Jolla, United States ; [‡] The Victor Chang Cardiac Research Institute, Sydney, Australia

[§]Deceased

**Competing interests:** The authors declare that no competing interests exist.

**Abstract** Interprotein electron transfer underpins the essential processes of life and relies on the formation of specific, yet transient protein-protein interactions. In biological systems, the detoxification of sulfite is catalyzed by the sulfite-oxidizing enzymes (SOEs), which interact with an electron acceptor for catalytic turnover. Here, we report the structural and functional analyses of the SOE SorT from *Sinorhizobium meliloti* and its cognate electron acceptor SorU. Kinetic and thermodynamic analyses of the SorT/SorU interaction show the complex is dynamic in solution, and that the proteins interact with $K_d = 13.5 \pm 0.8$ μM. The crystal structures of the oxidized SorT and SorU, both in isolation and in complex, reveal the interface to be remarkably electrostatic, with an unusually large number of direct hydrogen bonding interactions. The assembly of the complex is accompanied by an adjustment in the structure of SorU, and conformational sampling provides a mechanism for dissociation of the SorT/SorU assembly.

## Introduction

Although electron transfer reactions are key biochemical events, which underpin fundamental processes, such as respiration and photosynthesis, the study of the molecular details of the interprotein interactions at their core can be largely intractable. In particular, atomic resolution crystal structures of electron transfer complexes are rare, due to their fundamentally transient nature (*Antonyuk et al., 2013*). Electron transfer pathways are made up of chains of redox proteins, which provide a path for the controlled flow of electrons and rely on efficient docking of protein redox partners through noncovalent, dynamic protein-protein interfaces (*Moser et al., 1992*). Complementary electrostatic surfaces, hydrophobic interactions and dynamics at the protein-protein interface have all been proposed to contribute to efficient interprotein electron transfer (*Leys and Scrutton, 2004*), with a strong correlation between the driving force for the reaction, the distance between redox centers and the rate of electron transfer (*Moser et al., 1992; Marcus and Sutin, 1985*).

Interprotein electron transfer processes are central to the redox conversions of cellular sulfur compounds, which are an evolutionarily ancient type of metabolism that has existed as long as cellular life (*Schidlowski, 1979; Kappler et al., 2008*). Sulfur-containing compounds mediate many crucial reactions in the cell (for example, in coenzyme A, sulfur containing amino acids or glutathione), but their reactivity also makes them potentially toxic (*Kappler, 2011*). Sulfite in particular, is a highly reactive sulfur compound that can cause damage to proteins, DNA and lipids, resulting in oxidative

**eLife digest** A key feature of many important chemical reactions in cells is the transfer of particles called electrons from one molecule to another. The sulfite oxidizing enzymes (or SOEs) are a group of enzymes that are found in many organisms. These enzymes convert sulfite, which is a very reactive compound that can damage cells, into another compound called sulfate. As part of this process the SOE transfers electrons from sulfite to other molecules, such as oxygen or a protein called cytochrome c. In the past, researchers have described the three-dimensional structure of three SOEs using a technique called X-ray crystallography. However, it has been difficult to study how SOEs pass electrons to other molecules because of the temporary nature of the interactions.

McGrath et al. studied an SOE called SorT, which is found in bacteria. The SorT enzyme passes electrons from sulfite to another protein called SorU. McGrath used X-ray crystallography to determine the three-dimensional structures of versions of these proteins from a bacterium called *Sinorhizobium meliloti*. This included structures of the proteins on their own, and when they were bound to each other. These structures revealed that a subtle change in the shape of SorU occurs when the proteins interact, which enables an electron to be quickly transferred.

McGrath et al. also found that the interface between the two proteins showed an unexpectedly high number of contact sites. These strengthen the interaction between the two proteins, which helps to make electron transfer more efficient. However, these contact sites do not prevent the two proteins from quickly moving apart after the electrons have been transferred. The next challenge is to find out whether these observations are common to SOEs from other forms of life.

stress and irreversible cellular damage (*Feng et al., 2007*). In most cells, the detoxification of sulfite by oxidation to sulfate (*Equation 1*) is catalyzed by sulfite oxidizing enzymes (SOEs) (*Kisker et al., 1997*).

$$SO_3{}^{2-} + H_2O \longrightarrow SO_4{}^{2-} + 2H^+ + 2e^- \tag{1}$$

SOEs from plants, higher animals and bacteria have been characterized and they all catalyze the same fundamental reaction. However, their cellular functions, catalytic properties (*Kappler, 2011*; *Feng et al., 2007*; *Kappler and Wilson, 2009*; *Hille, 2002*; *Hänsch et al., 2007*) and the identities of their natural electron acceptors vary significantly. Some SOEs transfer electrons to oxygen (*Schrader et al., 2003*), while others interact with redox proteins such as cytochrome *c* (*Kisker et al., 1997*; *Kappler and Wilson, 2009*; *Low et al., 2011*; *Bailey et al., 2009*; *Cohen and Fridovich, 1971*; *Cohen and Fridovich, 1971*; *Cohen et al., 1971*) or as yet unknown cellular components (*Kappler, 2011*; *Low et al., 2011*; *Bailey et al., 2009*; *Wilson and Kappler, 2009*). To date, three unique crystal structures of SOEs have been reported: from chicken, plant and bacteria, which differ significantly in their domain architectures and redox cofactor compositions. However, none of these studies show details of a SOE in complex with its external electron acceptor (*Kisker et al., 1997*; *Schrader et al., 2003*; *Kappler and Bailey, 2005*). At present, no structural information on the molecular interactions of any of these enzymes with their respective electron acceptors is available and the determinants that dictate the type of electron acceptor individual SOEs employ, while maintaining the efficiency of the basic enzyme reaction are open questions. (*Kappler, 2011*; *Kappler, 2008*)

Here, we have investigated an electron transfer complex involving the periplasmic SorT sulfite dehydrogenase from the α-Proteobacterium *Sinorhizobium meliloti*, which represents a structurally uncharacterized type of SOE, and its electron acceptor, the *c*-type cytochrome SorU (*Low et al., 2011*; *Wilson and Kappler, 2009*). In *S. meliloti* the SorT sulfite dehydrogenase is part of a sulfite detoxification system that is induced in response to the degradation of sulfur containing substrates such as the organosulfonate taurine (*Wilson and Kappler, 2009*). Electrons derived from sulfite oxidation are passed on to the SorU cytochrome, and likely then to cytochrome oxidase, as *S. meliloti* is capable of sulfite respiration (*Low et al., 2011*). Here, we report the crystal structures of both the isolated SorT and SorU proteins and the biochemical and structural analyses of the SorT/SorU

electron transfer complex. This is the first time that a crystal structure of a molybdenum enzyme in complex with its external electron acceptor has been solved.

## Results

### The interactions between SorT and SorU are highly dynamic and efficient in solution

Sulfite-oxidizing enzymes, particularly those from bacteria, are known to be highly efficient catalysts (*Kappler, 2011*; *Kappler and Enemark, 2015*). Previous work has established that SorT is able to transfer electrons to the SorU cytochrome that is encoded on the same operon; however, no kinetic details of the interaction were reported (*Low et al., 2011*). With the artificial electron acceptor

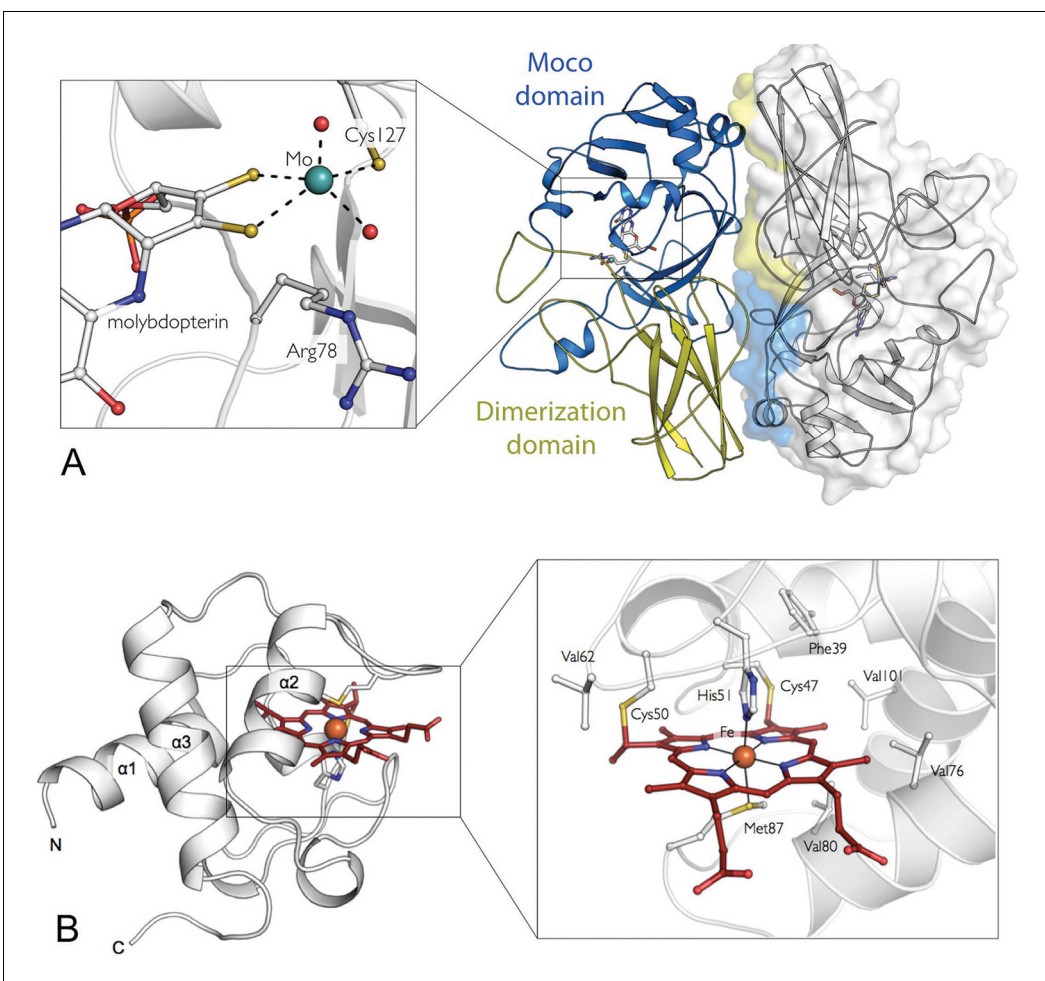

**Figure 1.** The crystal structures of the SorT and SorU proteins in isolation. (**A**) The structure of the SorT homodimer. Molecule A in blue/yellow; molecule B in gray (with transparent surface). For molecule A, the 'SUOX-fold domain' and 'dimerization domain' are represented in blue and yellow, respectively. The molybdopterin cofactor is shown as sticks within the SUOX-fold domain. The corresponding domains of the opposing protomer (shown in molecular surface representation), which constitute the dimer interface are colored to highlight the 'head-to-tail' dimer arrangement. INSET: a closer view of the molybdenum binding-site: the molybdenum atom (green sphere) is coordinated by two dithioline ligands from the molybdopterin (yellow spheres), residue Cys 127, an axial oxo ligand and an equatorial hydroxo or water ligand (red spheres). (**B**) The structure of SorU. The main three helices are labeled and the heme cofactor is shown in red. INSET: the heme binding site with the heme cofactor, coordinating residues, covalent links to Cys 50 and 57 and hydrophobic residues lining binding site: Phe 39, Val 62, Val 76, Val 80, Val 101 highlighted.

**Table 1.** Data collection and refinement statistics.

| | SorT | SorU | SorT/SorU complex |
|---|---|---|---|
| **Data collection** | | | |
| Space Group | $P2_1$ | $F222$ | $P2_12_12$ |
| Cell dimensions | | | |
| a, b, c (Å) | 96.0, 92.2, 109.4 | 70.9, 129.2, 197.0 | 109.6, 95.8, 49.9 |
| α, β, γ (°) | 90, 89.7, 90 | 90, 90, 90 | 90, 90, 90 |
| X-ray source | AUS MX2 | AUS MX2 | AUS MX2 |
| λ (Å) | 0.950 | 0.954 | 0.954 |
| Detector | ADSC Quantum 315r | ADSC Quantum 315r | ADSC Quantum 315r |
| Resolution range (Å) | 50-2.4 (2.43-2.35)[a] | 50-2.2 (2.28-2.20) | 50-2.5 (2.50-2.59) |
| Observed reflections | 240521 | 96340 | 64273 |
| Unique reflections | 77971 | 23453 | 18883 |
| Completeness (%) | 98.4 (99.4) | 99.9 (100) | 99.3 (99.6) |
| Multiplicity | 3.1 (3.1) | 4.1 (4.1) | 3.4 (3.4) |
| $<I/\sigma(I)>$ | 6.7 (2.1) | 8.9 (1.6) | 8.8 (1.7) |
| $R_{merge}$ (%)[b] | 15.9 (66.7) | 13.3 (76.6) | 13.9 (76.6) |
| **Refinement** | | | |
| Reflections in working set | 74024 | 22113 | 17781 |
| Reflections in test set | 3927 | 1201 | 960 |
| Protomers per ASU | 4 | 4 | 1 |
| Total atoms (non-H) | 11378 | 2941 | 3422 |
| Protein atoms | 10890 | 2542 | 3290 |
| Metal atoms | 4 | 4 | 2 |
| Water atoms | 376 | 227 | 63 |
| Other atoms | 108 | 168 | 67 |
| $R_{work}$ (%)[c] | 20.8 (31.7) | 19.2 (30.8) | 21.1 (30.2) |
| $R_{free}$ (%)[d] | 23.9 (34.7) | 24.0 (34.3) | 26.0 (36.5) |
| Rmsd bond lengths (Å) | 0.008 | 0.006 | 0.012 |
| Rmsd bond angles (deg) | 1.08 | 0.91 | 1.41 |
| $<B>$ (Å$^2$)[e] | 32.5 | 20.6 | 38.0 |
| Cruickshank's DPI | 0.07 | 0.23 | 0.49 |
| PDB ID | 4PW3 | 4PWA | 4PW9 |

[a]Values in parenthesis are for highest-resolution shell

[b] $R_{merge} = \sum_{hkl} \sum_i |I_i(hkl) - <I(hkl)>|/\sum_{hkl}\sum_i I_i(hkl)$

[c] $R_{work} = \sum_h |F_{obs} - F_{calc}|/\sum_h F_{obs}$

[d] Calculated as for $R_{work}$ using 10% of the diffraction data that had been excluded from the refinement

[e]As calculated by BAVERAGE (**Winn et al., 2011**)

ferricyanide, SorT was shown to have a turnover number of 338 ± 3 s$^{-1}$ (**Low et al., 2011**; **Wilson and Kappler, 2009**). Employing SorU as the substrate, we analyzed the kinetics of the interaction between SorT and SorU and found the interaction to be fast and highly specific, with a $K_M$ (SorU) of 32 ± 5 μM and a $k_{cat}$ of 140 ± 11 s$^{-1}$, confirming that SorU is the natural electron acceptor of SorT. Measurement of the thermodynamics of the SorT/SorU interaction by isothermal titration

calorimetry (ITC) revealed a dissociation constant of $K_d$ = 13.5 ± 0.8 µM with a determined stoichiometry of 0.8 ± 0.2. These values are in the range observed for other electron transfer complexes (*Dell'acqua et al., 2008*; *Pettigrew et al., 2003*) and match a model where SorT sequentially transfers two electrons, derived from sulfite oxidation, to two SorU molecules. In other words, the SorT/SorU complex must form twice (with two different ferric SorU protein molecules) to complete the oxidative half reaction of SorT.

The $K_M$ of SorT for SorU is very close to its affinity for sulfite ($K_M$ = 15.5 ± 1.9 µM [*Wilson and Kappler, 2009*]), and the turnover number in the SorU-based assay is ~40% of that seen with ferricyanide as the electron acceptor (*Wilson and Kappler, 2009*), where no significant reorientation and docking of the electron acceptor is required. These data reveal interesting details about the formation of the SorT/SorU complex, which appears to form with similar affinities between the two proteins when both are oxidized as in the ITC experiments and in a system where SorT constantly undergoes oxidation and reduction (SorU-based enzyme assay). In addition, the similarity between the determined values of $K_d$ and $K_M$ indicates that the affinity of SorT for SorU is unaffected by the presence of substrate or product. However, the kinetic parameters for this interaction are clearly distinct from those for other sulfite-oxidizing enzymes, such as the bacterial SorAB enzyme or chicken sulfite oxidase (CSO), both of which have much higher affinities for their respective electron accepting cytochromes $c$ (both with $K_{M(Cyt\ c)}$ ca. 2 µM) (*Kappler et al., 2006*). The catalytic turnover of CSO is relatively slow ($k_{cat}$ = 47.5 ± 1.9 s$^{-1}$), due to a mechanism requiring internal rearrangement, while turnover of SorAB with its natural electron acceptor (334 ± 11 s$^{-1}$) is significantly faster than that for SorT (*Kappler and Enemark, 2015*; *Kappler et al., 2006*).

## The crystal structure of the SorT homodimer reveals a head to tail subunit arrangement

In order to investigate whether there are any structural reasons for these differences, we solved the crystal structure of SorT by molecular replacement and refined it to 2.4 Å resolution. The structure shows two homodimeric assemblies per asymmetric unit (*Table 1*), which is in agreement with the quaternary structure as determined by MALLS (*Low et al., 2011*; *Wilson and Kappler, 2009*). Unexpectedly however, within the SorT homodimer the protomers are oriented in a head-to-tail orientation (*Figure 1A*), a subunit arrangement that has not been previously observed in structures of sulfite-oxidizing enzymes. In keeping with the nomenclature applied to other structurally-characterized SOEs, a 'dimerization' domain typically defines the interface between the two monomers (*Kisker et al., 1997*; *Schrader et al., 2003*), but the structure of the SorT dimer does not follow this paradigm.

Nevertheless, the fold of the SorT monomers is similar to those of other SOEs (*Kisker et al., 1997*; *Schrader et al., 2003*; *Kappler and Bailey, 2005*), comprising a central 'SUOX-fold' domain (*Workun et al., 2008*) that harbors the Mo active site and a 'dimerization' domain (*Figure 1A*). The SorT active site has a square-pyramidal geometry seen in all other SOE structures with a five coordinate molybdenum atom and a single tricyclic pyranopterin cofactor (*Figure 1A*, *Table 2*) (*George and Pickering, 1999*).

Single electron reduction of SorT to its EPR active Mo$^V$ form was achieved using a combination of Ti(III)citrate and a suite of organic redox mediators. The Mo$^V$ EPR spectrum is similar to the so-called 'high pH' EPR signature of SOEs (*Figure 2A*, *Table 3*, Appendix 1). An additional feature is the presence of superhyperfine coupling between the unpaired electron on the Mo ion and two nearby $I$ = ½

**Table 2.** Mo coordination geometry in the active site of SorT.

| Bond | Distance (Å) |
| --- | --- |
| Mo-S1 (pterin) | 2.4 |
| Mo-S2 (pterin) | 2.4 |
| Mo-S (Cys 127) | 2.3 |
| Mo=O | 1.7 |
| Mo-OH/H$_2$O | 1.9 |

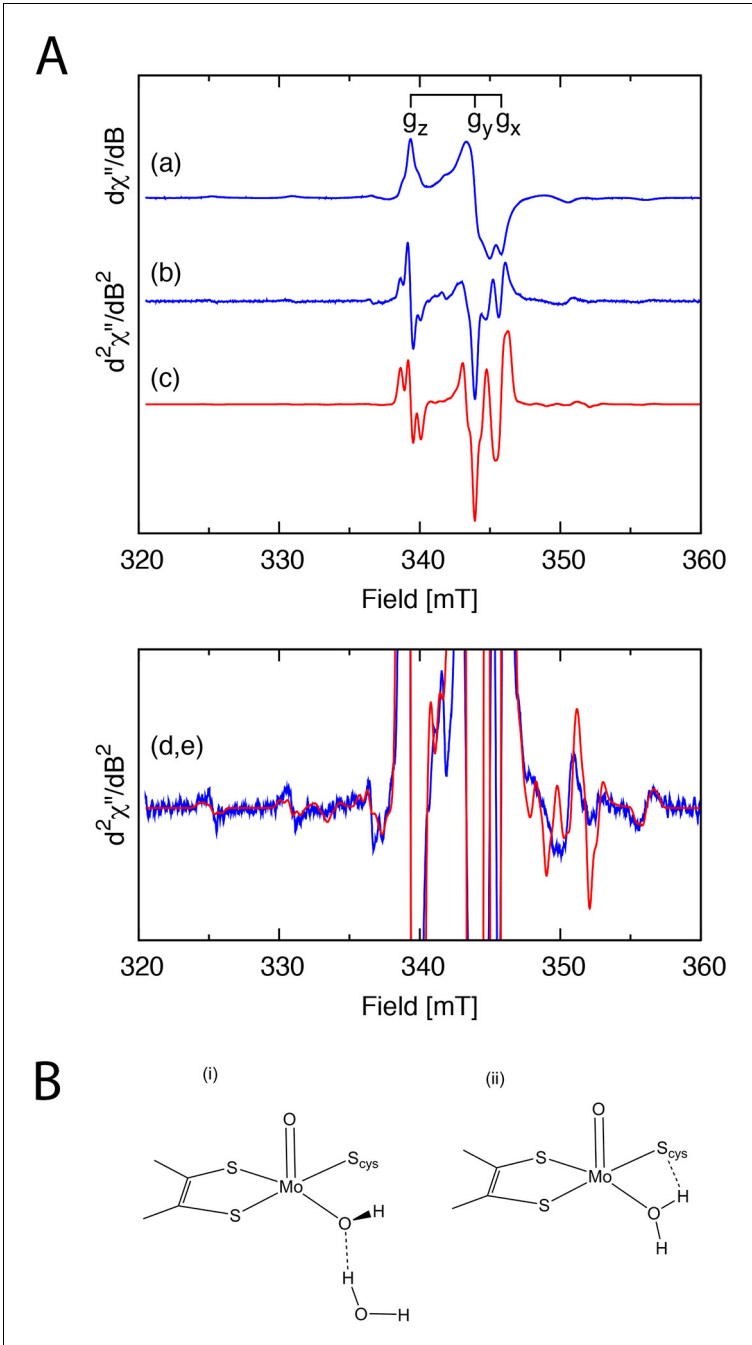

**Figure 2.** EPR analysis of the SorT protein. (**A**) X-band EPR spectra of the Mo(V) center in SorT. (a) First and (b) second derivative EPR spectra of SorT at 0 mV vs NHE in tricine pH 8.0, $\upsilon$= 9.43462 GHz, T = 136.3 K. (c) Computer simulation of the second derivative spectrum with the spin Hamiltonian parameters listed in *Table 3*; (d, e) Expansion of spectra (b) and (c), respectively. (**B**) Schematic structures of the (i) high and (ii) low pH forms of sulfite oxidase.

nuclei. This implies that the equatorially coordinated O-donor is an aqua ligand at pH 8 (*Figure 2B (ii)*) or that a hydroxido ligand is hydrogen bonded with a water molecule whose proximal H-atom is coupled with the electron spin on Mo (*Figure 2B(i)*, Appendix 1).

Within the SorT dimer, the subunit interface involves both the 'SUOX-fold' and the 'dimerization' domains (*Figure 1A*), resulting in a buried surface area of ~1280 Å$^2$ per monomer, which is

**Table 3.** Spin Hamiltonian parameters for the Mo(V) center of SorT and various low and high pH forms of human, avian, plant and bacterial sulfite oxidases.

| Species | Parameter | X | Y | Z | β° | Ref |
|---|---|---|---|---|---|---|
| SorT | g | 1.94930 | 1.95997 | 1.98632 | - | |
| | A($^{95}$Mo)[b] | 20.5 | 36.0 | 53.9 | 26 | |
| | A($^1$H)[b,c] | 3.5 | 4.0 | 4.8 | 0 | |
| SorA[c] | g | 1.9541 | 1.9661 | 1.9914 | -[d] | *Klein, et al., 2013* |
| Human SO | g Low pH | 1.9646 | 1.9723 | 2.0023 | - | *Enemark, et al., 2010* |
| | A($^1$H)[b] | 11.47 | 7.10 | 7.71 | - | *Enemark, et al., 2010* |
| Chicken SO | g (Low pH) | 1.9658 | 1.9720 | 2.0037 | - | *Drew and Hanson, 2009* |
| | A($^1$H)[b] | 11.93 | 7.37 | 7.95 | - | *Drew and Hanson, 2009* |
| | g (High pH) | 1.9531 | 1.9641 | 1.9872 | | *Drew and Hanson, 2009* |
| *A. Thaliana* SO | g (Low pH) | 1.963 | 1.974 | 2.005 | - | *Enemark, et al., 2006* |
| | A($^1$H)[b] | 11.9 | 9.2 | 10.3 | – | *Enemark, et al., 2006* |
| | g (High pH) | 1.956 | 1.964 | 1.989 | - | *Enemark, et al., 2006* |

[a]Non-coincident angle between g and A (rotation about x axis). [b]Units $10^{-4}$ cm$^{-1}$. [c] Two magnetically equivalent protons (I=1/2) were included in the computer simulated spectra. [c]$^{95}$Mo hyperfine couplings were unresolved and the shoulders on $g_z$ were incorrectly attributed to $^{95}$Mo hyperfine resonances. [d]Euler angles were not determined.

approximately 9% of the solvent-accessible surface of each monomer. The functional consequences and structural origins of these significant differences among the quaternary structures of SOEs are at this point unknown, as all of these enzymes are highly catalytically active, have similar active site structures and no known kinetic cooperativity that would imply a functional role for the different oligomeric assemblies (*Kappler and Wilson, 2009*; *Hänsch et al., 2007*; *Wilson and Rajagopalan, 2004*).

## In complex, SorT and SorU form a SorU/SorT₂/SorU assembly that reveals a pathway for electron transfer

Despite the dynamic nature of the SorT/SorU interaction, it was possible to co-crystallize SorT with SorU, resulting in the crystal structure of the SorT/SorU complex, where a single SorT/SorU entity is present per asymmetric unit, and the application of crystallographic 2-fold symmetry reveals a SorU/SorT₂/SorU assembly (*Figure 3A*, *Table 1*). Small-angle X-ray scattering (SAXS) with a sample prepared as a stoichiometric mixture of SorT and SorU (*Figure 3B*, *Table 4*, *Appendix 2-Figure 1*) confirmed that the structure observed in the crystal is preserved in solution.

The central assembly within the SorU/SorT₂/SorU complex is the SorT homodimer, which is identical to the structure of the SorT homodimer alone (*Figures 1A* and *3A*, *Table 1*). The structure of the SorU protein, both within the SorT/SorU complex and when crystallized in isolation (*Figure 1B*, *Table 1*), is predominantly α-helical with three major α-helices arranged to form a bundle that frames the heme-binding site (*Figure 1B*) ($C_{47}XXC_{50}H$, axial ligands: His 51 and Met 87).

In the SorT/SorU complex, the SorU protein docks within a pocket adjacent to the SorT active site, with the heme cofactor located at the protein-protein interface (*Figures 3A* and *4A*). The shortest 'edge-to-edge' distance between the SorT Mo atom and the propionate group from the SorU heme *c* cofactor is 8.2 Å, which is well within the distance for fast electron transfer through the protein medium (*Page et al., 1999*). PATHWAY analysis (*Onuchic et al., 1992*) (*Table 5*) further indicates that the dominant electron tunneling pathway from SorT to SorU proceeds from the Mo atom, via the coordinating $H_2O/OH^-$, to the guanidinium group of SorT residue Arg 78 and across the protein-protein interface to the heme propionate group and to the pyrrole ring of heme *c* to the heme iron (*Figure 4B*, *Table 5*).

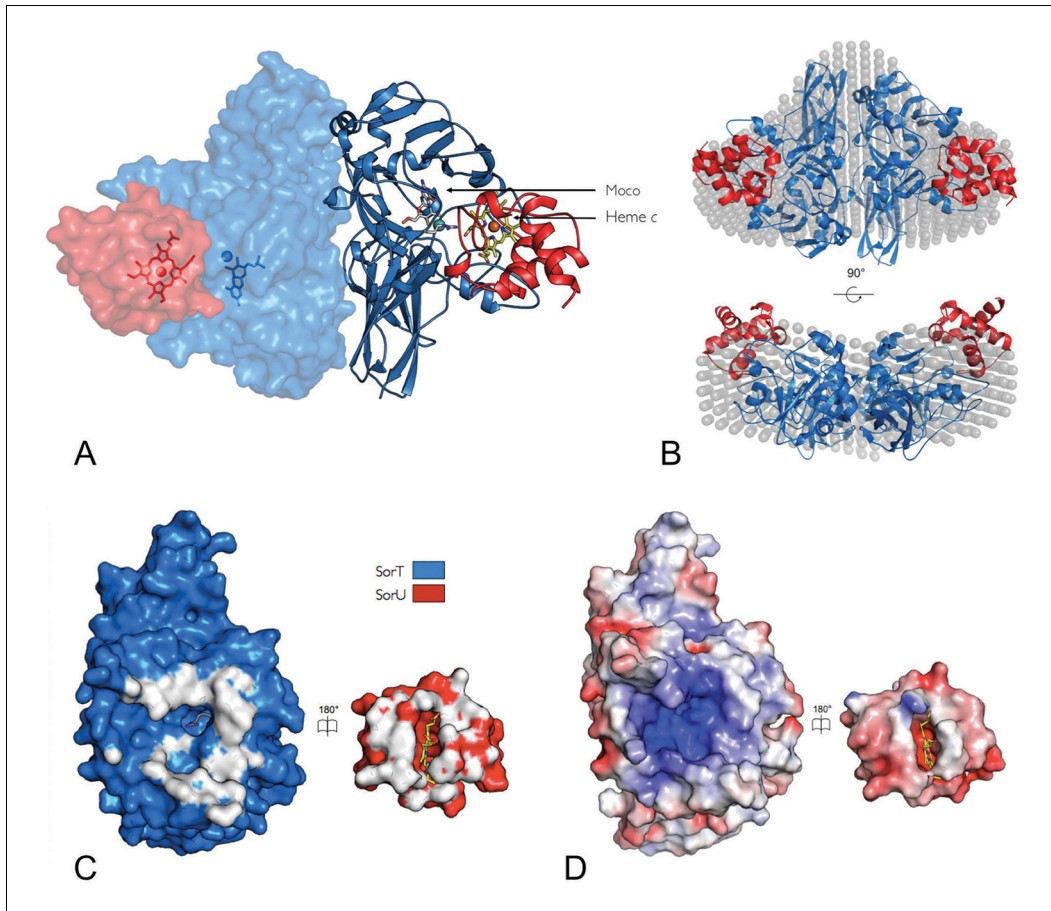

**Figure 3.** The structure of the SorT/SorU electron transfer complex. (**A**) The asymmetric unit from the crystal structure of the SorT/SorU complex contains the functional electron transfer complex. The SorU/SorT$_2$/SorU complex is revealed by the application of crystallographic symmetry operators. The positions of the redox active molybdenum (SorT) and heme *c* (SorU) cofactors are indicated. (**B**) Two views of an overlay of the SorU/SorT$_2$/SorU crystal structure with the averaged and filtered dummy atom model from 10 *ab initio* reconstructions as revealed by SAXS analyses. (**C**) 'Open-book unfolding' of SorT/SorU complex (SorT is shown in blue, SorU in red) indicating the 'footprint' of interfacing residues from each protein. (**D**) The same view as Panel C, showing the charge complementarity of the SorT/SorU interface (areas of positive charge in blue, negative charge in red and neutral in white).

## Conformational change reminiscent of an 'induced fit' mechanism facilitates docking and electron transfer between SorT and SorU

Specific structural adaptations of the SorU protein take place when the SorT/SorU assembly is formed (*Figure 5B*). A surface loop on SorU (residues 82–93) moves away from the SorT/SorU complex interface, leading to a reorientation of the heme ligand residue Met 87 so that a different Met 87 rotamer coordinates the iron in the SorU structures within and outside of the complex (*Figure 5B*). This change in the structure of SorU is required to allow a Mo-heme edge-to-edge distance of 'closest approach' of ca. 8 Å within the SorT/SorU assembly. Without this adjustment (for example, if the SorU residue 82–93 loop structure remained rigid) the closest approach for the redox cofactors would be ca. 10 Å.

Changes in the conformation of axial heme ligands are known to alter the orbital interactions between the iron atom and the ligand, which can change the redox properties of the heme group (*Tai et al., 2013*). However, this does not appear to be the case here as the redox potential of the SorU heme was determined by optical spectroelectrochemistry to be +108(±10) and +111(±10) mV vs. NHE (pH 8.0), respectively, in the presence and absence of SorT (*Table 6*, *Figure 6B*).

**Table 4.** Data collection and processing parameters for analysis of the SorT/SorU complex in solution by Small Angle X-ray Scattering (SAXS).

| Data collection parameters | |
|---|---|
| Instrument | SAXSess (Anton Paar) |
| Beam geometry | 10 mm slit |
| AH, LH (Å$^{-1}$), GNOM beam geometry definition | 0.28, 0.12 |
| $q$-range measured (Å$^{-1}$) | 0.01-0.400 |
| Exposure time (min) | 60 (4 x 15) |
| SorT$_2$SorU$_2$ concentration range (mg mL$^{-1}$) | 2.75-5.5 |
| Temperature (°C) | 10 |
| **Structural parameters***| |
| $R_g$ (Å), $I(0)$ (cm$^{-1}$) from Guinier (desmeared data) $q*R_g < 1.3$ | 30.8 ± 0.4, 0.223 ± 0.002 |
| $R_g$ (Å), $I(0)$ (cm$^{-1}$) from $P(r)$ ($q$-range 0.01 – 0.25 Å$^{-1}$) | 32.0 ± 0.3, 0.235 ± 0.002 |
| $d_{max}$ (Å) from $P(r)$ | 110 |
| *Molecular mass determination* * | |
| Molecular mass $M_r$ from Guinier $I(0)$ (ratio with expected) | 108741 (0.984) |
| Molecular mass $M_r$ from $P(r)$ $I(0)$ (ratio with expected) | 114593 (1.037) |
| **SorT$_2$SorU$_2$ parameters calculated from sequence and chemical composition** | |
| Molecular volume (Å$^3$) | 134385 |
| Molecular weight $M_r$ (Da) | 110556 |
| Partial specific volume (cm$^3$ g$^{-1}$) | 0.732 |
| Contrast (X-rays) ($\Delta\rho$ x 10$^{10}$ cm$^{-2}$) | 2.895 |
| **Modeling results and validation** | |
| Crystal structure $R_g$, $d_{max}$ (Å) SorT/SorU$_2$/SorT SorT | 31.3, 108 27.9, 99 |
| Crystal structure compare to desmeared $I(q)$ ($\chi$-value) SorT/SorU$_2$/SorT ($q$-range 0.01 – 0.15 Å$^{-1}$) SorT ($q$-range 0.01 – 0.15 Å$^{-1}$) | 1.7 2.3 |
| Results from 10 *ab initio* shape restorations. P1 symmetry: Average molecular volume (Å$^3$) Normalised spatial distribution (NSD) and NSD variation $\chi$ value for fit to desmeared data | 140800 0.508 (0.008) 1.8 |
| **Software employed** | |
| Calculation of expected $M_r$, $\triangle\rho$ and $\upsilon$ values | MULCh |
| Primary data reduction, $I(q)$ vs $q$ | SAXSQuant 1D |
| Desmearing | SAXSQuant |
| Guinier analysis | PRIMUS |
| $P(r)$ analysis | GNOM |
| Model $I(q)$ from crystal coordinates | CRYSOL |
| *ab initio* shape restorations | DAMMIN |
| 3D graphics representations | PYMOL |

*Reported for 2.75 mg ml$^{-1}$ measurement.

The Mo$^{VI/V}$ and Mo$^{V/IV}$ redox potentials of SorT (+110(±10) mV and -18(±10) mV vs. NHE (pH 8)) were determined by an EPR-monitored redox titration where the initial EPR-silent Mo$^{VI}$ form is reduced to the EPR-active Mo$^V$ state (**Figure 2A**), which then gives way to the EPR-silent Mo$^{IV}$ form

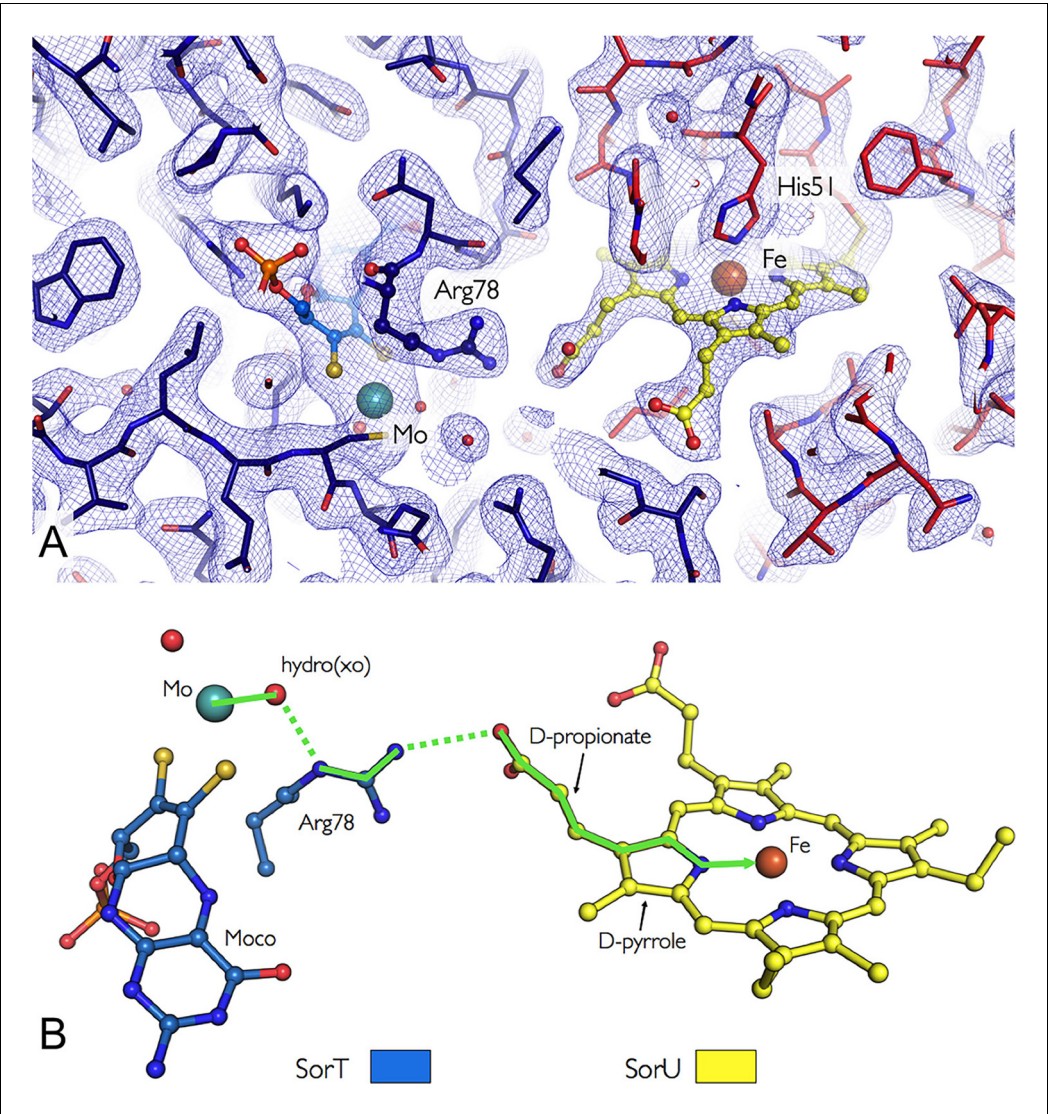

**Figure 4.** Orientation of the redox cofactors in the crystal structure of the SorT/SorU electron transfer complex. (A) Electron density map in the region of the SorT/SorU interface. The SorT molecule is represented in blue and the SorU molecule in red. The 2Fo-Fc electron density map (contoured at 1σ) is shown as a blue net and the redox cofactors (molybdenum and heme) are colored according to the representation in Panel **B**. (B) Pathway for electron transfer (*Beratan et al., 1992*).

at low potential, resulting in a bell-shaped curve (*Table 6*, *Figure 6A*). The high $Mo^{VI/V}$ potential at pH 8 matches that of the ferris/ferrous redox couple. (*3,4*).

The structural change in SorU indicates that an 'induced fit' mechanism is responsible for the formation of a productive SorT/SorU electron transfer complex. This type of mechanism has in the past been used to describe electron transfer complexes (involving electron transfer flavoproteins or ferredoxin reductases) where the protein partners include mobile domains, and where conformational change is necessary for the creation of high affinity protein-protein interfaces (*Senda et al., 2007*; *Toogood et al., 2007*). However, although facilitating redox interactions, these systems are distinct from the docking mechanism seen for the SorT and SorU proteins which accompanies modifications to the structure of SorU and allows the two redox centers to attain positions of closest approach for fast electron transfer.

**Table 5.** Electron transfer parameters between SorT (Mo) and SorU (Fe) as calculated by PATHWAYS (*Onuchic et al., 1992*).

| | |
|---|---|
| Distance (Mo-Fe, Å) | 16.5 Å |
| Atomic packing density ($\rho$) | 0.97 |
| Average decay exponential ($\beta$) | 0.97 |
| Electronic coupling ($H_{DA}$) | $3.4 \times 10^{-4}$ |
| Maximum ET rate (s$^{-1}$) | $1.2 \times 10^{7}$ |

## The SorT/SorU interface features extensive electrostatic interactions

The SorT/SorU complex interface shows significant charge complementarity, with the negative charge on SorU correlating with a concentration of positive charges at the SorU binding site on SorT (*Figures 3C,D*). Unlike other known cytochromes *c* that can act as electron acceptors to SOEs

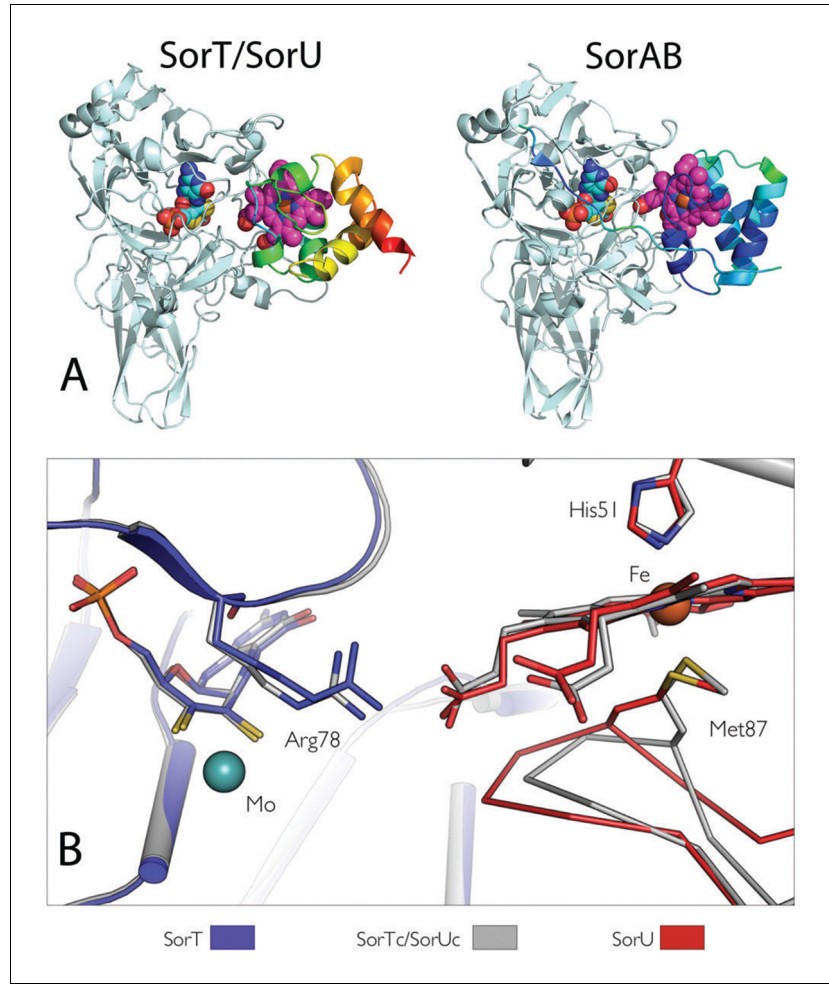

**Figure 5.** Comparisons of (**A**) the SorT/SorU and SorAB structures and (**B**) the structures of SorT and SorU within and outside of the electron transfer complex. (**A**) Structures of the SorT/SorU (left) and SorAB (right) complexes, where the Cα traces of the heme-containing protomers are colored according to temperature factor. (**B**) Superposition of the SorT and SorU structures within and outside of the electron transfer complex, highlighting conformational changes that were observed to accompany complex formation. The crystal structures of SorT and SorU within the SorT/SorU complex are shown in gray, and the superposed structures of SorT and SorU determined alone are shown in blue and red respectively.

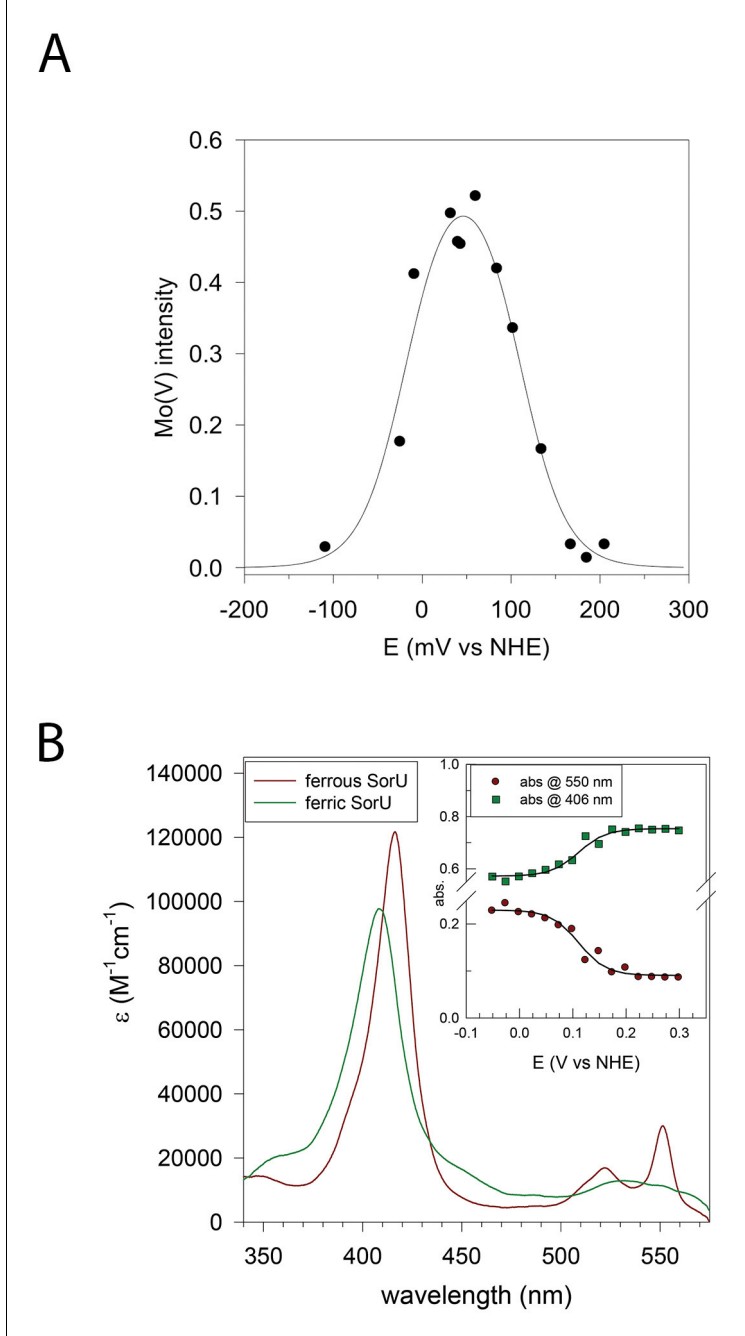

**Figure 6.** Redox analyses of the SorT protein and the SorT/SorU complex. (**A**) Plot of EPR intensity ($I_p$) at 343 mT (from Mo$^V$ form of SorT) as a function of solution redox potential (E mV vs NHE). The solid line is a fit to the equation $I(E) = \frac{I_p}{1 = 10^{(E-E_1)/59} + 10^{(E_2-E)/59}}$ using the potentials E$_1$ = Mo$^{VI/V}$ = +110(±10) mV and E$_2$ Mo$^{V/IV}$-18(±10) mV vs NHE). (**B**) Electronic spectra of ferric and ferrous SorU obtained from spectroelectrochemistry. Inset: plot of absorbance at 550 nm (ferrous α-band) and 406 nm (ferric Soret band) as a function of applied potential. The solid lines are theoretical curves based on the equation $Abs = \frac{(\varepsilon_{ox}10^{(E-E')/59} + \varepsilon_{red})}{1 + 10^{(E-E')/59}} C_{tot}$ where the extinction coefficients refer to the oxidized and reduced forms of the protein and *Abs* is the absorbance at this same wavelength. $c_{tot}$ is the total protein concentration. The redox potential (*E'* = +111 mV vs NHE) was obtained by global analysis of all potential dependent spectra across all wavelengths with the program ReactLab Redox (Maeder and King).

**Table 6.** Redox potential values for SorT and SorU[a].

| Protein | Couple | E° (mV vs NHE) |
|---|---|---|
| SorT | Mo$^{VI/V}$ | +110(±10) |
| | Mo$^{V/IV}$ | -18(±10) |
| SorU | Fe$^{III/II}$ | +108 (±10) |
| SorU (in the presence of SorT) | Fe$^{III/II}$ | +111 (±10) |

[a]Redox potentials of SorT were determined by redox potentiometry, and SorU redox potentials by optical spectroelectrochemistry.

(*Low et al., 2011*; *Brody and Hille, 1999*; *Kappler et al., 2000*), the electrostatic surface of SorU has an overall negative charge (*Figures 3C,D*). The positive charge on the SorT surface therefore explains the low catalytic activity of SorT with horse heart cytochrome *c* (7 U/mg, pI ~10), the natural electron acceptor for vertebrate SOEs, compared to the high activity observed with SorU (212 U/mg; pI ~4) (*Kappler, 2011*; *Low et al., 2011*; *Wilson and Kappler, 2009*). In its electrostatic nature, the interaction surface between SorT and SorU is unusual. Structures of other cytochrome-containing electron transfer complexes (*Nojiri et al., 2009*; *Axelrod et al., 2002*; *Pelletier and Kraut, 1992*; *Solmaz and Hunte, 2008*) show binding interfaces characterized by a 'ring' of electrostatic interactions that encompass contact surfaces that are predominantly hydrophobic. In fact, the 'steering' of electron transfer partners by electrostatic interactions, accompanied by 'tuning' *via* hydrophobic interactions is a dominant observation for protein-protein electron transfer complexes (*Nojiri et al., 2009*; *Axelrod et al., 2002*; *Pelletier and Kraut, 1992*; *Solmaz and Hunte, 2008*).

## An unusually large number of hydrogen bonds and salt bridges characterize the SorT/SorU interface

In addition to the electrostatic interactions that support the formation of the SorT/SorU complex, there are six hydrogen bonds found at the SorT/SorU protein-protein interface, as well as a salt bridge between the SorT active site residue Arg 78 and a propionate group of the SorU heme moiety (*Table 7*, *Figure 7*). This is an unusually large number in comparison with structures of other cytochrome-containing electron transfer complexes (*Nojiri et al., 2009*; *Axelrod et al., 2002*; *Pelletier and Kraut, 1992*; *Solmaz and Hunte, 2008*), which tend to have fewer hydrogen bonds and lack salt bridges. In fact, direct hydrogen bonds between electron transfer proteins are generally considered unfavorable for a transient interaction because of energetically disadvantageous

**Table 7.** Comparison of the protein-protein interfaces in the SorT/SorU and SorAB structures.

| Parameter | SorT/SorU[a] | | SorAB[b] | |
|---|---|---|---|---|
| | SorT | SorU | SorA | SorB |
| Average relative *B* factor[c] (Å$^2$) | 0.9 | 1.5 | 1.0 | 1.1 |
| Buried surface area (Å$^2$)[d] | 644 | 696 | 1254 | 1380 |
| Interfacing residues[d] | 31 | 21 | 46 | 33 |
| Hydrogen-bonds | 6 | | 30[e] | |
| Salt-bridges | 1 | | 2[e] | |
| Shape complementarity statistic[f] | 0.63 | | 0.77 | |

[a]This work

[b]PDB code 2BLF (*Kappler and Bailey, 2005*)

[c]Calculated as the average for the protomer of interest divided by the average for the entire complex structure.

[d](*Krissinel and Henrick, 2007*)

[e]Taken from(*Kappler and Bailey, 2005*)

[f](*Lawrence and Colman, 1993*)

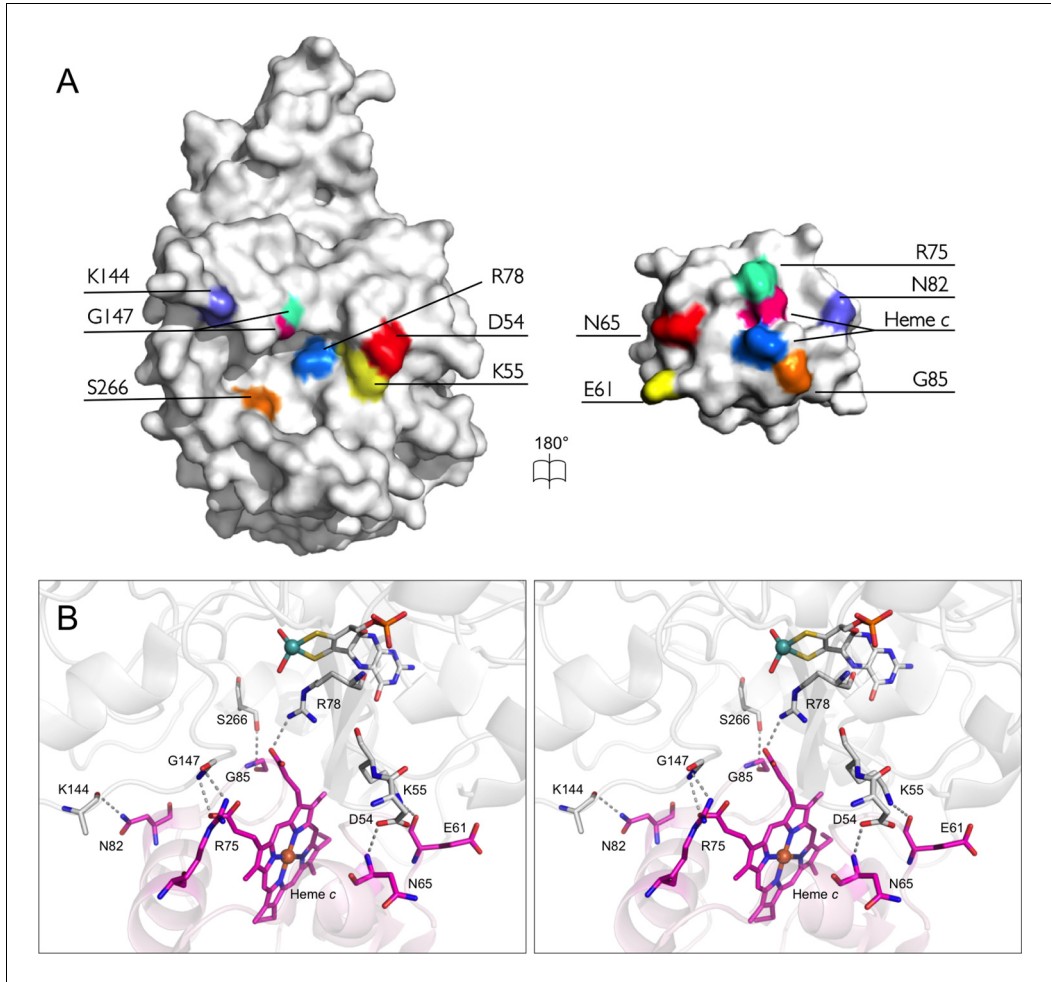

**Figure 7.** The bonding network at the interface of SorT/SorU. (**A**) An open-book representation depicts residues involved in forming stable bonds at the interface between SorT and SorU as corresponding color patches mapped onto the molecular surface. (**B**) Stereoview of the interface between SorT and SorU. Bonding residues are shown as sticks with bonds shown as dashes between atoms. SorT is shown in light grey and SorU is shown in magenta.

desolvation (*Miyashita et al., 2003*). Also significant is the observation that no intermolecular interactions at the interface are mediated by water molecules (*Figure 7*) (*Nojiri et al., 2009*; *Gray and Winkler, 2005*). In fact, very little ordered water is observed in proximity to the interfacing region of the SorT molecule (a total of 2 water molecules only and these are hydrogen bonded to the SorT molecule rather than mediating the SorT/SorU interaction). However, the current analysis is limited by the moderate resolution of the current structure (2.6 Å), which as a consequence, includes only ca. 0.15 modeled water molecules per residue.

Compared with the subunit interface of the SorAB bacterial sulfite dehydrogenase, which contains 30 hydrogen-bonds and 2 salt bridges (*Kappler and Bailey, 2005*), supporting a permanent, heterodimeric complex of a heme *c* subunit (SorB) and a Mo cofactor containing (SorA) catalytic subunit (*Kappler and Bailey, 2005*) (*Table 7*), the extent of the subunit interactions in the SorT/SorU structure is modest. The difference between the permanent SorAB and the transient SorT/SorU complexes is also illustrated by calculations of the shape complementarity and the buried surface areas between the protomers (*Lawrence and Colman, 1993*), with the latter being about twice as large for the SorAB assembly than for SorT/SorU (*Table 7*). Interestingly, much of the additional contact area between molecules in the SorAB structure derives from the SorB N-terminal structure (residues B501-B518, PDB 2BLF), which extends away from the core of the subunit, wraps around the SorA

'SUOX-fold' domain and contributes one salt-bridge and 6 hydrogen bonding interactions (*Table 7*) (*Kappler and Bailey, 2005*). This feature is absent from the SorU structure.

## The dynamic SorT/SorU interaction observed in solution is reflected in the crystal

Despite the intricate assembly of interactions at the protein-protein interface, but in agreement with the kinetics and thermodynamics of the SorT/SorU interaction in solution, the SorT/SorU contact is dynamic, as illustrated by a temperature factor analysis of the complex structure. Within the SorT/SorU complex, the SorU protein shows a significantly increased average atomic temperature factor (reflecting significant flexibility) relative to the structure of the SorT protomer (1.5 versus 0.9 $\text{Å}^2$, respectively; *Table 7*). Furthermore, the relative temperature factors per residue for the SorU molecule increase with increasing distance from the SorT/SorU interface (*Figure 5A*), indicating that the SorU molecule is dynamic relative to SorT within the crystalline lattice. In contrast, the 'static' SorAB complex shows uniform, low temperature factors for both redox subunits (*Table 7*, *Figure 5A*). This observation is an exquisite illustration of 'conformational sampling' within the SorT/SorU electron transfer complex, which results from the conformational flexibility of one protein redox partner relative to the other and both facilitates electron exchange by accessing the optimal orientations of each redox partner and promotes fast dissociation of the complex following transfer (*Leys and Scrutton, 2004*; *van Amsterdam et al., 2002*).

## Discussion

By describing the structure of the SorT/SorU complex in this work, we report the first example of a structure of an SOE in complex with its external electron acceptor; all previous structures of SOEs being of the enzymes and their internal heme domains or subunits only (*Kisker et al., 1997*; *Schrader et al., 2003*; *Kappler and Bailey, 2005*). The structure of the SorT/SorU complex therefore allows insights into electron transfer in what is thought to be a highly prevalent type of bacterial SOE (*Low et al., 2011*) and into protein-protein electron transfer in general. While the complex shows dynamic adaptations similar to those demonstrated previously for electron transfer complexes and has a dissociation constant of the right order of magnitude, it also shows some features that have not been seen in electron transfer complexes, namely an interface that is stabilized by a relatively large number of hydrogen bonds and salt bridges, and an induced fit docking mechanism.

The structure of the protein-protein interface in the SorT/SorU structure is particularly intriguing. The relative lack of bound water molecules, mirrors more the observations made of permanent heterodimeric complexes than transient interactions (*Gray and Winkler, 2005*). Previous investigations into the factors that influence protein-protein docking for electron transfer have shown that the strength of the protein-protein interaction correlates linearly with the product of the total charges on the protein partners (*Trana et al., 2012*; *Xiong et al., 2010*). In this way, the affinity between the SorT/SorU proteins ($K_d$ = 13.5 ± 0.8 μM) correlates with the fast measured turnover rates ($k_{cat}$ of 140 ± 11 s$^{-1}$) and with the predominantly electrostatic nature of the protein-protein interface.

The turnover number for SorT with SorU as the electron acceptor is also in the range of values measured for the human sulfite oxidase (HSO) and CSO (25.0 ± 1.3 s$^{-1}$ and 47.5 ± 1.9 s$^{-1}$, respectively), but significantly slower than that observed for the permanent SorAB complex (345 ± 11 s$^{-1}$) (*Kappler et al., 2006*; *Wilson and Rajagopalan, 2004*; *Brody and Hille, 1999*). Importantly, in the structure of SorAB, the docking site of the heme subunit (SorB) with the SorA subunit is almost identically positioned to that seen for SorT/SorU, with major differences existing only in the number of hydrogen bonds and salt bridges in the protein-protein interface. For the CSO and HSO enzymes, docking of the mobile heme *b* domain near the Mo active site has been proposed to be similar to that seen for SorT/SorU (*Utesch and Mroginski, 2010*). It should be noted, however, that the docking events in CSO (and HSO) and between SorT/SorU serve fundamentally different purposes: for CSO and HSO, domain docking enables intramolecular electron transfer and involves a heme domain that is an intrinsic part of the enzyme. This is a step that precedes interactions with the external electron acceptor for these enzymes. In contrast, and despite the fact that it is occupying a similar docking site to that predicted for CSO and HSO (*Utesch and Mroginski, 2010*), SorU is the external electron acceptor for SorT and the electron transfer is intermolecular.

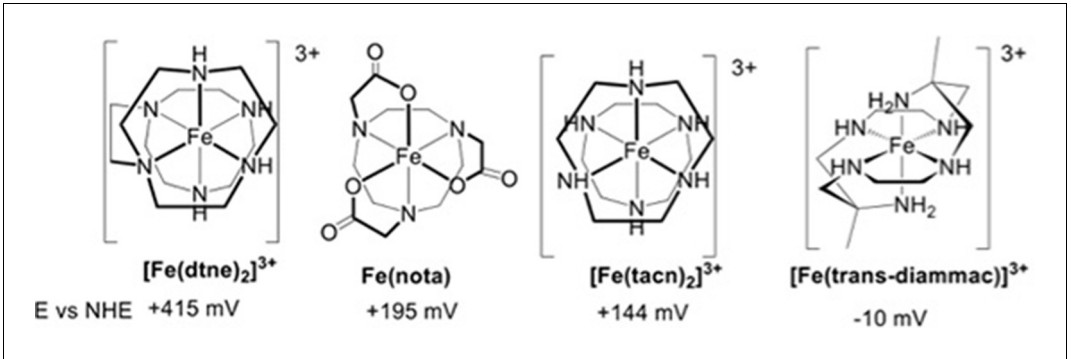

**Figure 8.** Redox mediators employed in optical spectroelectrochemistry experiments.

The SorT/SorU complex described here thus represents an elegant compromise between the requirements for fast and efficient electron transfer and reaction specificity. It also illustrates new aspects for highly dynamic protein-protein interactions: (i) A relatively large number of hydrogen bonds and salt bridges may be required to form the initial stable protein complex, but this does not preclude a dynamic protein – protein interaction; (ii) relatively subtle structural adjustments in one redox partner (SorU) can facilitate electron transfer by ideally locating the redox active cofactors in close proximity; (iii) the comparatively complex binding interface in SorT/SorU can be counterbalanced by the conformational sampling of one protein relative to the other, which enables the rapid dissolution of the complex following electron exchange.

It remains to be seen whether these principles apply to other SOE – external electron acceptor interactions. Future work should focus on investigating interaction interfaces in currently little studied SOEs where new types of interactions may be present as for many of these enzymes currently no external electron acceptor is known.

## Materials and methods

### Protein overexpression, purification, data collection and structure solution

Recombinant SorT and SorU proteins were overproduced and purified as previously described (*Low et al., 2011*), with minor modifications. SorT was crystallized by hanging drop vapor diffusion with drops consisting of equal volumes (2 µL) of protein and crystallization solution (0.1 M HEPES pH 7.5, 8% ethylene glycol, 0.1 M manganese (II) chloride tetrahydrate and 17.5% PEG 10,000) at 20°C. Crystals were cryoprotected in reservoir solution with 30% glycerol before flash-cooling in liquid nitrogen. Small (ca. 20 × 10 × 10 µ) crystals of SorU were grown in drops containing equal volumes (2 µL) of protein and reservoir solution (1.8 M tri-sodium citrate, pH 5.5, 0.1 M glycine), which were harvested and flash-cooled in liquid nitrogen without additional cryoprotection. Purified SorT (20 mM Tris pH 7.8, 2.5% glycerol) and SorU (20 mM Tris pH 7.8, 150 mM NaCl) were mixed and incubated on ice at a molar ratio of 2:1 (SorU:SorT; total protein concentration 8 mgmL$^{-1}$) before crystallization via hanging-drop vapor diffusion with a reservoir solution containing 0.2 M sodium formate, 0.1 M Bis-Tris propane pH 7.5 and 20% PEG 3350. Crystals grew to a maximum size of ca. 150 x 100 x 20 µ in 4 days at 20°C and were flash-cooled in liquid nitrogen after brief soaking in mother liquor containing 30% glycerol. All diffraction data were collected on an ADSC Quantum 315r detector at the Australian Synchrotron on beamline MX2 at 100 K and were processed with HKL2000 (*Otwinowski and Minor, 1997*). Unit cell parameters and data collection statistics are presented in *Table 1*.

The crystal structure of SorT was solved by molecular replacement using PHASER (*McCoy et al., 2007*) with a search model generated with CHAINSAW (*Chainsaw, 2008*) from the SorA portion of the SorAB crystal structure (29.0% sequence identity, Protein Data Bank entry 2BLF [*Kappler and Bailey, 2005*]) as a template (*Larkin et al., 2007*). The resulting model was refined by iterative cycles of amplitude based twin refinement (using twin operators H, K, L and –H, -K, L with estimated twin

fractions of 0.495 and 0.505 respectively) within REFMAC (*Murshudov et al., 2011*), interspersed with manual inspection and correction against calculated electron density maps using COOT (*Emsley and Coot, 2004*). The refinement of the model converged with residuals $R$ = 0.208 and $R_{free}$ = 0.239 (*Table 1*). The structure of the SorT/SorU complex was solved by molecular replacement using PHASER (*McCoy et al., 2007*), with the refined SorT structure as a search model. Initial rounds of refinement yielded a difference Fourier electron density map, which clearly showed positive difference density for the location of one molecule of SorU per asymmetric unit, which was manually built using COOT (*Emsley and Coot, 2004*). Refinement was carried out with REFMAC5 (*Murshudov et al., 2011*) and PHENIX (*Adams et al., 2002*) and converged with residuals $R$ = 0.211 and $R_{free}$ = 0.260 (*Table 1*). The refined SorU model, from the SorT/SorU complex structure, was used as a search model to solve the SorU structure by molecular replacement using PHASER (*McCoy et al., 2007*). Refinement was carried out with REFMAC5 and PHENIX (*Adams et al., 2002*) and converged with residuals $R$= 0.192 and $R_{free}$ = 0.240. All structures were judged to have excellent geometry as determined by MOLPROBITY (*Chen et al., 2010*)(*Table 1*).

## Small-angle X-ray scattering (SAXS)

SAXS analysis of the SorT/SorU complex was performed in a buffer of 20 mM Tris pH 7.8, 2.5% v/v glycerol. Purified SorU and SorT were mixed and incubated on ice at a molar ratio of 2:1 (SorU: SorT), generating two samples of total protein concentrations 2.75 and 6.25 mgmL$^{-1}$, respectively. SAXS data were measured as described previously(*Jeffries et al., 2011*) with the data collection parameters listed in *Table 4*. Data were reduced to $I(q)$ vs $q$ ($q=4\pi sin\theta\lambda$,where $q=4sin2\theta$ is the scattering angle) using the program SAXSquant that includes corrections for sample absorbance, detector sensitivity, and the slit geometry of the instrument. Intensities were placed on an absolute scale using the known scattering from H$_2$O. Protein scattering was obtained by subtraction of the scattering from the matched solvents (20 mM Tris pH 7.8, 2.5% v/v glycerol obtained from the flow-through after protein concentration by centrifugal ultrafiltration). Molecular weight ($M_r$) estimates for the proteins were made using the equation from Orthaber(*Orthaber et al., 2000*): $Mr=N_A I(0)C\triangle\rho M2$ where $N_A$is Avogadro's number, $C$ is the protein concentration and $\triangle\rho M=\triangle\rho\upsilon$, where $\triangle\rho$ is the protein contrast and $\upsilon$ the partial specific volume, both of which were determined using the program MULCh(*Whitten et al., 2008*).

The ATSAS program package(*Volkov and Svergun, 2003*) was used for data analysis and modeling, with the specific programs used detailed in *Table 4*, along with the data ranges and results of each of the calculations. Further detail on data interpretation and analysis for these experiments is detailed in Appendix 2.

## Electron paramagnetic resonance (EPR) spectroscopy

Continuous-wave X-band (ca. 9 GHz) (CW) electron paramagnetic resonance (EPR) spectra were recorded with a Bruker Elexsys E580 CW/pulsed EPR spectrometer fitted with a super high Q resonator; the microwave frequency and magnetic field were calibrated with a Bruker microwave frequency counter and a Bruker ER 036TM Teslameter, respectively. A microwave power of 20 mW was used and optimal spectral resolution was obtained by keeping the modulation amplitude to a 1/10 of the linewidth. A flow-through cryostat in conjunction with a Eurotherm (B-VT-2000) variable temperature controller provided temperatures of 127–133 K at the sample position in the cavity.

Bruker's Xepr (version 2.6b.45) software was used to control the data acquisition including, spectrometer tuning, signal averaging, temperature control and visualization of the spectra. Computer simulation of the EPR spectra were performed with the following spin Hamiltonian (*Equation 2*)

$$H = \beta B \cdot g \cdot S + S \cdot A\left(^{95,97}Mo\right)\cdot I - g_n\beta B\cdot I + \sum_{i=1}^{2}(S\cdot A(^1H)\cdot I - g_n\beta_n B\cdot I) \qquad (2)$$

using the XSophe-Sophe-XeprView (version 1.1.4) computer simulation software suite(*Hanson et al., 2004*; *Hanson et al., 2013*) on a personal computer, running the Mandriva Linux v2010.2 operating system. Further detail on data interpretation and analysis for these experiments is detailed in Appendix 1.

## EPR-monitored redox potentiometry

The $Mo^{IV/V}$ and $Mo^{V/VI}$ redox potentials of SorT were determined by an EPR-monitored redox titration carried out in a Belle technology anaerobic box. The protein solution (1.5 mL, 40-90 µM in Tris-HCl, pH 8.0 and 10% glycerol) also contained the following redox mediators at concentrations of ~50 µM: diaminodurol (2,3,5,6-tetramethylphenylene-1,4-diamine, $E_{m,7}$ +276 mV), dichlorophenolindophenol ($E_{m,7}$ +217 mV), 2,6-dimethylbenzoquinone ($E_{m,7}$ +180 mV), phenazine methosulfate ($E_{m,7}$ +80 mV), 2,5-dihydroxybenzoquinone ($E_{m,7}$ –60 mV) indigo trisulfonate ($E_{m,7}$ -90 mV), 2-hydroxy-1,4-naphthoquinone ($E_{m,7}$ -152 mV) and anthraquinone 2-sulfonate ($E_{m,7}$ -230 mV). The reductant was Ti (III) citrate and the oxidant was $NaS_2O_8$ (both ~100 mM). After addition of titrant and equilibration (15–30 min), the equilibrium potential was measured with a combination Pt wire/AgAgCl redox electrode attached to a Hanna 8417 meter calibrated against the quinhydrone redox couple ($E^{o'}$ (pH 7) = +284 mV vs NHE). A 100 µL aliquot of protein was withdrawn and transferred to an EPR tube (in the anaerobic box) which was then sealed and then carefully frozen in liquid nitrogen (outside the box). Potentials for all experiments were measured with a combination Pt wire-Ag/AgCl electrode attached to a Hanna 8417 meter. The intensity of the $Mo^V$ signal ($I$) was recorded as a function of measured potential ($E$), *Figure 6*.

## Optical spectroelectrochemistry

Spectroelectrochemistry of SorU in isolation and the SorT:SorU complex was performed with a Bioanalytical Systems BAS100B/W potentiostat connected to a Bioanalytical Systems thin layer spectroelectrochemical cell (0.5 or 1 mm pathlength) bearing a transparent Au mini-grid working electrode, a Pt wire counter and Ag/AgCl reference electrode. Redox mediators (all Fe complexes) used in the experiment were employed at concentrations of 50 µM (*Figure 8*).

None exhibit any significant absorption in the spectral range of interest at micromolar concentrations. The total solution volume was ca. 700 µL. The buffer was 20 mM Tris (pH 8) containing 200 mM NaCl as supporting electrolyte. The SorU concentration was ca. 50 µM, while experiments on the SorU:SorT complex used approximately equal concentrations of both proteins (50 µM). Spectra were acquired within a Belle Technology anaerobic box with an Ocean Optics USB2000 fibre optic spectrophotometer. Initially, the cell potential was poised at ca. -100 mV and the system was allowed to equilibrate until no further spectral changes were apparent (fully reduced SorU). The potentials were then increased in 50 mV increments and the spectrum was measured when no further changes were seen (5–10 min). When the protein was fully oxidized the potential was scanned in the reverse direction in 50 mV intervals to establish reversibility. Data were fitted using the program ReactLab Redox (Maeder and King).

## Isothermal titration calorimetry (ITC)

Affinity measurements were conducted using a Microcal ITC200 system (GE Healthcare) at 25°C using SorT and SorU in buffer (25 mM HEPES pH 7.5, 150 mM NaCl and 2.5% glycerol) at final concentrations of 300 µM and 30 µM, respectively. SorT at a concentration of 300 µM was titrated with eighteen injections (2.0 µl each) of SorU. All affinity measurements were performed in triplicate and fitted using a single site mode. Protein concentrations were estimated using Bicinchoninic acid (BCA) protein assay kit (Thermo Fisher Scientific, Waltham, MA).

## Enzyme kinetics

Sulfite dehydrogenase enzyme assays were carried out as described previously (*Low et al., 2011*; *Wilson and Kappler, 2009*; *Kappler et al., 2000*). The reduced – oxidized extinction coefficient for SorU at 550 nm was 17.486 $mM^{-1}$ $cm^{-1}$ as determined by spectroelectrochemistry. Data fitting was carried out using Sigmaplot 12 (Systat).

## Acknowledgements

This work was supported by a La Trobe Institute for Molecular Science (LIMS) Senior Research Fellowship to MJM, an Australian Research Council grant and Fellowship (DP0878525) to UK and an NHMRC CJ Martin Postdoctoral Research Fellowship to APM. PVB also acknowledges financial support from the Australian Research Council (DP1211465). Aspects of this research were undertaken

on the Macromolecular Crystallography beamline at the Australian Synchrotron (Victoria, Australia) and we thank the beamline staff for their enthusiastic and professional support. A/Prof Matthew Perugini (La Trobe University) is thanked for very helpful discussions. Assistance with EPR septcroscopy from Dr Jeff Harmer and Dr Chris Noble (Centre for Advanced Imaging, University of Queensland) is also gratefully acknowledged.

## Additional information

### Funding

| Funder | Grant reference number | Author |
|---|---|---|
| Australian Research Council | Discovery Project Grant DP0878525 | Ulrike Kappler |
| National Health and Medical Research Council | CJ Martin Postdoctoral Research Fellowship 1053624 | Aaron P McGrath |
| Australian Research Council | Discovery Project Grant DP1211465 | Paul V Bernhardt |

The funders had no role in study design, data collection and interpretation, or the decision to submit the work for publication.

### Author contributions

APM, JT, PVB, GRH, Acquisition of data, Analysis and interpretation of data, Drafting or revising the article; ELL, GPCG, MK, BC, Acquisition of data, Analysis and interpretation of data; JMG, Analysis and interpretation of data, Drafting or revising the article; UK, MJM, Conception and design, Acquisition of data, Analysis and interpretation of data, Drafting or revising the article

## Additional files

### Major datasets

The following datasets were generated:

| Author(s) | Year | Dataset title | Dataset URL | Database, license, and accessibility information |
|---|---|---|---|---|
| McGrath AP, Maher MJ | 2015 | Crystal structure of the sulfite dehydrogenase SorT from Sinorhizobium meliloti | http://www.rcsb.org/pdb/explore/explore.do?structureId=4PW3 | Publicly available at the Protien Data Bank (accession no. 4PW3) |
| Laming EM, McGrath AP, Maher MJ | 2015 | Crystal structure of the c-type cytochrome SorU from Sinorhizobium meliloti | http://www.rcsb.org/pdb/explore/explore.do?structureId=4PWA | Publicly available at the Protien Data Bank (accession no. 4PWA) |
| McGrath AP, Maher MJ | 2015 | Crystal structure of the electron-transfer complex formed between a sulfite dehydrogenase and a c-type cytochrome from Sinorhizobium meliloti | http://www.rcsb.org/pdb/explore/explore.do?structureId=4PW9 | Publicly available at the Protien Data Bank (accession no. 4PW9) |

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

## Appendix 1

### Supplemental material on the interpretation of EPR data

The optimum potential to measure the continuous wave (CW) EPR spectrum of the Mo(V) center in SorT was found to be +50 mV from the redox potentiometry experiments (*Figure 6A*). The CW EPR spectrum at +0 mV (*Figure 2A(a)*) arises from a single rhombically distorted Mo(V) center.

Naturally abundant Mo consists of a mixture of isotopes ($^{95,97}$Mo, I=5/2, 25.5% abundance; $^{92,94,96,98,100}$Mo, I=0, 74.5% abundance) and, consequently, the spectrum consists of three I=0 resonances corresponding to the principal directions of the g-matrix and six (2I+1) satellite resonances, each with an intensity of ~4%. Increased spectral resolution was obtained by numerically differentiating the spectrum and carefully Fourier filtering (Hamming function) the spectrum to remove the high-frequency noise without distorting the spectrum (*Figure 2A(b)*). Interestingly, a close examination of the $g_z$ resonance around 339.35 mT reveals a triplet (*Figure 2A*) (*Hanson et al., 2004*; *2013*) in an approximate ratio of 1:2:1. Computer simulation of the CW EPR spectrum with a monoclinic ($C_s$ symmetry) spin Hamiltonian (*Equation 2*) incorporating two magnetically equivalent I=1/2 nuclei and the spin Hamiltonian parameters listed in *Table 3* produces the spectrum shown in *Figure 2A(c)*.

Previously we have shown in a study of model Mo(V) complexes that the non-coincident angle, $\beta$ (rotation of $A_{y,z}$ from $g_{y,z}$ about x) can be accurately determined utilizing multifrequency CW EPR in conjunction with computer simulation studies (*Drew et al., 2007a*; *2007b*). Herein, the increased spectral resolution in the second derivative spectrum and the narrow line widths enable an accurate determination of the non-coincident angle $\beta$ without resorting to the use of multiple microwave frequencies. Computer simulation of the second derivative spectrum showed that the $^{95,97}$Mo hyperfine resonant field positions along the 'z' and 'y' directions were highly sensitive to the magnitude of these hyperfine couplings and the non-coincident angle $\beta$. The excellent agreement between the simulated and experimental second derivative spectra (*Figure 2A(b)(c)*) gives confidence in the values of the spin Hamiltonian parameters. We have also shown through a systematic density functional theory study that the non-coincident angle $\beta$ can be correlated to the pterin fold angle (*Drew and Hanson, 2009*), and for $\beta$=26°, the predicted pterin fold angle would be 5.6°, which is in good agreement with that determined (1.9°±0.2°) from the X-ray crystal structure of SorT.

CW and pulsed EPR spectra of the Mo(V) center in human, avian, plant and bacterial sulfite have been extensively studied (*Table 3*)(*Kappler et al., 2000*; *Enemark et al., 2010*; *Doonan et al., 2008*; *Lamy et al., 1980*; *Astashkin et al., 2002*; *Klein et al., 2013*; *Enemark et al., 2006*; *Astashkin et al., 2005*). The Mo(V) CW EPR spectra of SO from humans, birds and plants are pH dependent. The CW EPR spectra arise from a Mo(V) center with rhombic or lower symmetry and at low pH the resonances are split into a doublet arising from a strongly coupled proton (*Table 3*). CW and pulsed EPR spectra in conjunction with isotope enrichment ($^{17}$O, $^2$H) studies have identified the origin of the proton as an equatorial hydroxo ligand. Loss of the proton superhyperfine coupling at high pH in the CW EPR spectrum, was shown through orientation selective multifrequency electron spin echo envelope modulation (ESEEM) studies to be a rotation of the hydroxo moiety out of the xy plane (rather than deprotonation), thereby reducing the unpaired electron spin density on the $^1$H nucleus of the hydroxo ligand (*Enemark et al., 2010*; *Klein et al., 2013*; *Astashkin et al., 2009*). $^{17}$O ESEEM studies also revealed two other weakly coupled oxygen ligands, the axially coordinated Mo=O moiety and a weakly coupled OH$^-$ moiety hydrogen bonded to the equatorial hydroxyo ligand (*Klein et al., 2013*; *Astashkin et al., 2009*).

In contrast, the Mo(V) EPR spectra of SOEs from bacteria (SorA [*Kappler et al., 2000*] and SorT) are pH independent. A comparison of the $g_z$ resonances from SorA(*Kappler et al.,*

2000) and SorT (Figure 2) shows their lineshapes to be very similar, both exhibiting shoulders. Herein, we have clearly shown through computer simulation of the Mo(V) CW EPR spectrum that the shoulders do not arise from $^{95,97}$Mo hyperfine coupling (Figure 2, Table 3), but arise from two weakly coupled $^1$H nuclei. While the computer simulation assumed that the two protons were magnetically equivalent, a slight magnetic inequivalence of the two $^1$H superhhyperfine couplings cannot be ruled out. The origin of the $^1$H superhyperfine couplings is likely to be either (i) an equatorial aqua ligand with the protons lying outside of the equatorial plane, thereby reducing the overlap with Mo based $d_{xy}$ orbital containing the unpaired electron or (ii) an equatorial hydroxo ligand which is hydrogen bonded to another hydroxyl moiety (Figure 2B) where both protons lie outside of the equatorial plane. The proton superhyperfine couplings are similar to those found from pulsed EPR studies of SorA (Enemark et al., 2010; Raitsimring et al., 2005).

## Appendix 2

### Supplemental material for the interpretation of SAXS data

The scattering data from the sample prepared as a stoichiometric mixture of SorT and SorU is presented in *Figure 3B* and *Appendix 2-figure 1*, while *Table 4* summarizes the structural parameters derived from those data. Significantly the $M_r$ value determined from $I(0)$ is in excellent agreement with that expected for a 2:2 complex (within 2-4% for calculations based on Guinier or $P(r)$ derived $I(0)$). Further, the scattering data fit the crystal structure of the SorT/SorU complex reasonably well, yielding a $\chi$ value of 1.7 for the overall fit and good agreement with the crystal structure $R_g$ value. This agreement is not expected to be perfect as the crystal structure is missing a total of 56 residues: 33 from the N-terminus of the SorT, 2 from the N-terminus of SorU and 21 from the C-terminus of SorU; accounting for 12% of the total molecular mass. By comparison, a significantly worse $\chi$ value of 2.3 is obtained by fitting only the SorT dimer, which also predicts a significantly smaller $R_g$ value than was observed (28 Å compared to 31 Å).

Dummy atom reconstructions yielded shapes that had the expected molecular volume and in projection had the expected shape for the $SorU/SorT_2/SorU$ assembly, although the shapes were consistently a somewhat flatter disk-like structure than that observed in the crystal structure (*Figure 3B*). This result is not unexpected, as this class of structure (flattened anisotropic particles) is known to present some difficulty on *ab initio* shape reconstructions (*Volkov and Svergun, 2003*).

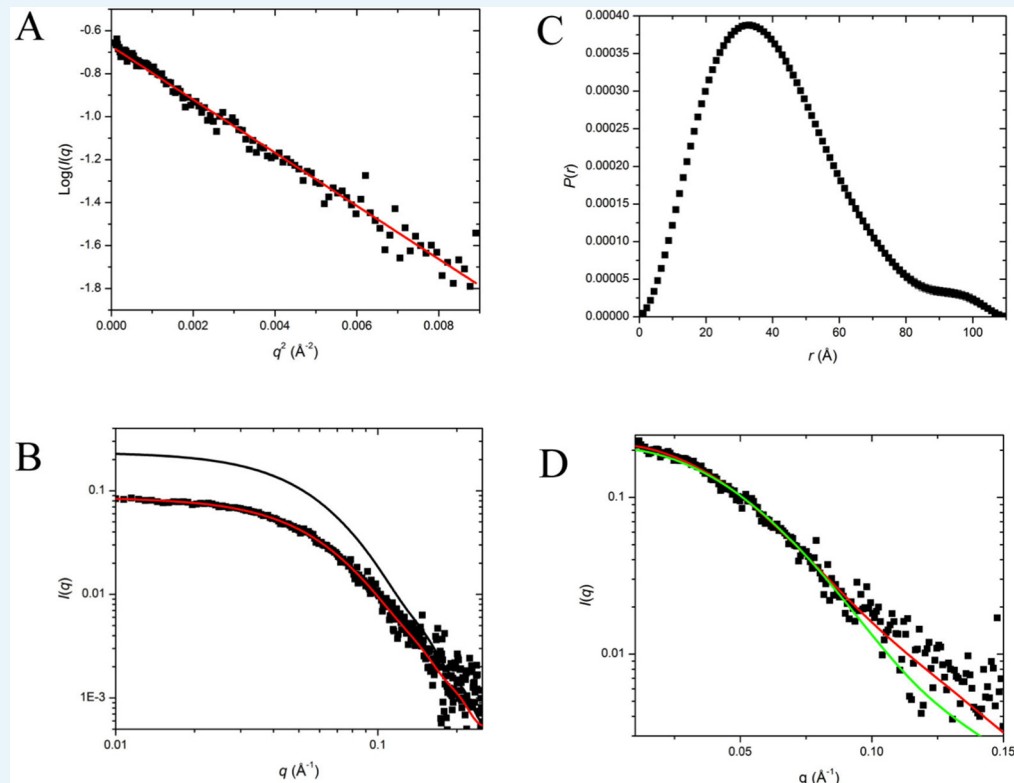

**Appendix 2-figure 1.** SAXS Data and Interpretation.
(**A**) Guinier plot of the desmeared $I(q)$ versus $q$; (**B**) log:log plot of the measured (slit smeared) $I(q)$ versus $q$ with the $P(r)$- model $I(q)$ (black line) and smeared $P(r)$-model $I(q)$ (red line) fits; (**C**)

$P(r)$ versus $r$ for the $P(r)$ model in (B), $d_{max}$ is 110 Å; (**D**) superposition of the desmeared $I(q)$ versus $q$ with that calculated from the crystal structure of the SorT/SorU$_2$/SorT complex (red line) and the SorT dimer (green line).

