## [Decision Letter]

Thank you for submitting your work entitled "Structural basis of interprotein electron transfer in bacterial sulfite oxidation" for peer review at *eLife*. Your submission has been favorably evaluated by Michael Marletta (Senior editor). Michael also served as the Reviewing editor, and sought advice from two reviewers.

The reviewers have discussed the reviews with one another and Michael Marletta has drafted this decision to help you prepare a revised submission.

Summary:

This manuscript by McGrath et al. reports the structure of a sulfite oxidizing enzyme (SorT) and its interaction with an electron acceptor SorU, a c-type cytochrome. The authors make the point of the importance of sulfite oxidizing enzyme (SOEs) in overall sulfur metabolism. They note this work is the first co-structure of an SOE with its native electron acceptor, and they further note this is likely the first such characterization for a Mo-containing enzyme. They compare and contrast their SorT structure to that of other SOE structures. In the co-structure they characterize the possible electron transfer pathway between the Mo of SorT and the heme of the SorU. Conformational changes of SorU that allow this interaction to shorten in the co-protein structure vs rigid docking of the individual monomers is described. The protein electrochemistry shows no significant change in the midpoint potential of SorU upon interacting with SorT. The authors discuss the electrostatic interactions between the two proteins, which they consider unusual for cytochrome based ET complexes, including the presence of salt bridges and 5 H-bonds. They then contrast to a relevant "permanent" heterodimer complex between SorB and SorA by which the SorU/SorT interactions are much lower, and the authors discuss specific differences in the protein/protein interactions of the two different pairs. The contrast is extended to looking at the relative temperature factors between proteins of the two pairs, with higher conformational flexibility in the SorU of the SorT/SorT complex. The high conformational flexibility is related by the authors to the thermodynamics of binding and the kinetics of the SorT reaction.

Essential revisions:

We feel that the focus of the paper is not on the protein-protein interface as we were led to believe. In fact there is no structure of the H bonding interactions (the R/propionate interaction is shown) and no discussion of water molecules at the interface. The authors need to either rewrite the paper to focus on the as advertised protein-protein interaction or refocus on a comparison of this sulfite dehydrogenase with others that have the heme domain tethered to the dehydrogenase domain.

The authors have done a lot of work and there are not many examples of two proteins that are involved in electron transfer. The work is thus interesting and important. However, there is little discussion of the well characterized proteins with not only structure, but kinetics and where electrostatic interactions have been changed and the kinetic consequences have been evaluated. The work of Brian Hoffman in particular, but also that of Crane and Gray, is relevant. The authors need to define the focus of the paper. Most of the Discussion, for example, was focused on evolution. The Introduction was too general and did not contain enough background information as provided on sulfite dehydrogenases and what is known.

Minor points:

1) The Abstract is way too general and needs details on the system being investigated, the *K*_d_ determined for the protein-protein interactions and the details of the interface. What are the oxidation states, for example of the crystallized proteins etc? What is the reaction catalyzed by SorT?

2) The Introduction is also way too general, ex about sulfur metabolism, and should be focused on the systems that have been well studied crystallographically and kinetically and what has been learned.

3) Figure 2 was very difficult to read due to poor choice of font. We could not really see the EPR spectra.

4) Figure 3 should be removed altogether as the reader does not see much and the description in the text is sufficient.

5) A new figure with the interface and the H bonding interactions, waters should be added in addition to Figure 6 with the salt bridge.

6) We think you should consider moving the details about the protein-protein interaction, such as *K*_d_ and turnover numbers, first rather than last.

7) Is the catch phrase "caught in the act" necessary?

[Editors' note: further revisions were requested prior to acceptance, as described below.]

Thank you for resubmitting your work entitled "Structural basis of interprotein electron transfer in bacterial sulfite oxidation" for further consideration at *eLife*. Your revised article has been favorably evaluated by Michael Marletta (Senior editor and Reviewing editor) and two reviewers. The manuscript has been improved but there are some remaining issues that need to be addressed before acceptance, as outlined below:

In the subsection title “The crystal structure of the SorT homodimer reveals a unique subunit arrangement”, the “unique subunit arrangement” is with respect to what? Why not say a unique head to tail orientation or something alike?

We found the structures in Figure 2 on the EPR figure confusing. How do you know there is a hydroxide ion?

---

## [Author Response]

*Essential revisions:*

*We feel that the focus of the paper is not on the protein-protein interface as we were led to believe. In fact there is no structure of the H bonding interactions (the R/propionate interaction is shown) and no discussion of water molecules at the interface. The authors need to either rewrite the paper to focus on the as advertised protein-protein interaction or refocus on a comparison of this sulfite dehydrogenase with others that have the heme domain tethered to the dehydrogenase domain. The authors have done a lot of work and there are not many examples of two proteins that are involved in electron transfer. The work is thus interesting and important. However, there is little discussion of the well characterized proteins with not only structure, but kinetics and where electrostatic interactions have been changed and the kinetic consequences have been evaluated. The work of Brian Hoffman in particular, but also that of Crane and Gray, is relevant. The authors need to define the focus of the paper. Most of the Discussion, for example, was focused on evolution. The Introduction was too general and did not contain enough background information as provided on sulfite dehydrogenases and what is known.*

We have included Figure 7, which illustrates the inter-protein hydrogen bonding interactions in the SorT/SorU complex. We note that the current resolution of the SorT/SorU (2.6 Å) structure limits the analysis of the water structure (with only 0.15 water molecules per residue modeled into the structure), but thank the reviewers for this suggestion. We have added a description within the Results section of the manuscript (subheading “An unusually large number of hydrogen bonds and salt bridges characterize the SorT/SorU interface”) to detail this part of the structure.

We have chosen to focus on the SorT/SorU protein-protein interaction as suggested and as described above included an additional figure and text to emphasise this part of the analysis.

We have used the work of Hoffman and Gray to guide our analyses as suggested (Onuchic et al., 1992; Gray and Winkler, 2005; Trana et al., 2012; Xiong et al., 2010 and Beratan et al., 1992).

We have refocused the Discussion away from the evolutionary aspects to a more detailed analysis of the structure in the context with the kinetics and thermodynamic measurements as suggested.

We have re-written the Introduction to provide more detail on the background to sulfite dehydrogenases in particular.

*Minor points:*

*1) The Abstract is way too general and needs details on the system being investigated, the* K_d_
*determined for the protein-protein interactions and the details of the interface. What are the oxidation states, for example of the crystallized proteins etc.? What is the reaction catalyzed by SorT?*

The Abstract has been entirely rewritten to include the suggested points.

*2) The Introduction is also way too general, ex about sulfur metabolism, and should be focused on the systems that have been well studied crystallographically and kinetically and what has been learned.*

As mentioned in ‘Essential Revisions’ above, the Introduction has been rewritten to include responses to these suggestions.

3) Figure 2 was very difficult to read due to poor choice of font. We could not really see the EPR spectra.

A new Figure 2 (now Figure 2 and Figure 6) with greater clarity has been included. We are happy to make further alterations to these figures if the current resolution remains insufficient.

*4) Figure 3 should be removed altogether as the reader does not see much and the description in the text is sufficient.*

We have removed Figure 3.

*5) A new figure with the interface and the H bonding interactions, waters should be added in addition to Figure 6 with the salt bridge.*

We have prepared a new Figure 7, which illustrates the hydrogen bonding structure at the protein-protein interface.

*6) We think you should consider moving the details about the protein-protein interaction, such as* K*_d_ and turnover numbers, first rather than last.*

These experiments are now detailed first in the Results section as suggested (subheading “The interactions between SorT and SorU are highly dynamic and efficient in solution”).

*7) Is the catch phrase "caught in the act" necessary?*

No; this has been removed.

[Editors' note: further revisions were requested prior to acceptance, as described below.]

*In the subsection title “The crystal structure of the SorT homodimer reveals a unique subunit arrangement”, the “unique subunit arrangement” is with respect to what? Why not say a unique head to tail orientation or something alike?*

We have changed the title of this section to the following as suggested: “The crystal structure of the SorT homodimer reveals a head to tail subunit arrangement”.

*We found the structures in Figure 2 on the EPR figure confusing. How do you know there is a hydroxideion?*

We thank the editor for pointing this out – there was an overlooked error in Figure 2, which has now been corrected. The figure now reflects the text on paragraph three, subheading “The crystal structure of the SorT homodimer reveals a head to tail subunit arrangement”.